# Novel Antibiofilm Inhibitor Ginkgetin as an Antibacterial Synergist against *Escherichia coli*

**DOI:** 10.3390/ijms23158809

**Published:** 2022-08-08

**Authors:** Yubin Bai, Weiwei Wang, Mengyan Shi, Xiaojuan Wei, Xuzheng Zhou, Bing Li, Jiyu Zhang

**Affiliations:** 1Key Laboratory of New Animal Drug Project of Gansu Province, Lanzhou 730050, China; 2Key Laboratory of Veterinary Pharmaceutical Development, Ministry of Agriculture, Lanzhou 730050, China; 3Lanzhou Institute of Husbandry and Pharmaceutical Sciences, Chinese Academy of Agricultural Sciences, Lanzhou 730050, China

**Keywords:** ginkgetin, *Escherichia coli*, antibiofilm, *EPS*, motility, quorum sensing, antibacterial synergist

## Abstract

As an opportunistic pathogen, *Escherichia coli* (*E. coli)* forms biofilm that increases the virulence of bacteria and antibiotic resistance, posing a serious threat to human and animal health. Recently, ginkgetin (Gin) has been discovered to have antiinflammatory, antioxidant, and antitumor properties. In the present study, we evaluated the antibiofilm and antibacterial synergist of Gin against *E. coli*. Additionally, Alamar Blue assay combined with confocal laser scanning microscope (CLSM) and crystal violet (CV) staining was used to evaluate the effect of antibiofilm and antibacterial synergist against *E. coli*. Results showed that Gin reduces biofilm formation, exopolysaccharide (*EPS*) production, and motility against *E. coli* without limiting its growth and metabolic activity. Furthermore, we identified the inhibitory effect of Gin on AI-2 signaling molecule production, which showed apparent anti-quorum sensing (QS) properties. The qRT-PCR also indicated that Gin reduced the transcription of curli-related genes (*csgA*, *csgD*), flagella-formation genes (*flhC*, *flhD*, *fliC*, *fliM*), and QS-related genes (*luxS*, *lsrB*, *lsrK*, *lsrR*). Moreover, Gin showed obvious antibacterial synergism to overcome antibiotic resistance in *E. coli* with marketed antibiotics, including gentamicin, colistin B, and colistin E. These results suggested the potent antibiofilm and novel antibacterial synergist effect of Gin for treating *E. coli* infections.

## 1. Introduction

Due to the overuse of antibiotics, and bacterial mutations, multidrug-resistant bacteria have emerged as one of the world’s most significant health problems today [1,2]. The National Institutes of Health (NIH) estimates that more than 16 million people die annually because of bacterial infections worldwide [3,4]. In addition, about 80% of them are linked to bacterial biofilm formation-mediated resistance [4]. The biofilm is an aggregate of microorganisms where microorganisms are adhered to the substrate and encapsulated within a self-produced matrix of *EPS*, proteins, and extracellular DNA (eDNA) [5,6,7]. It can enhance bacterial tolerance to antibiotics and innate immune response, ensuring survival and reproduction when antibiotic therapy stops [8]. In addition, biofilms contribute to adaptive mutations that may lead to antibiotic resistance [9]. Finally, the process associated with horizontal gene transfer increases in biofilm, increasing the range of plasmid stability and mobile genetic elements [10,11]. Biofilm-forming bacteria are less sensitive to antibiotics than planktonic cells once formed [12,13]. In turn, this has increased biofilm-producing bacteria’s resistance to antimicrobial agents and reduced their effectiveness in treating biofilm-associated infections [14]. A study of biofilm bacteria found that they could tolerate more than 1000 times as many antibiotics as planktonic bacteria [15]. Therefore, the biofilm has been considered a potential target for treating and managing clinical microbial infections, which has become an obvious drug target [16,17,18]. Combined with marketed antibiotics or alone, biofilm inhibitors reduce bacterial adhesion, thus reducing the persistence of infection, eliminating biofilm infection [19], and reducing the chance of drug-resistance development [20]. For example, combining L-arginine with gentamicin can effectively enhance the sensitivity of planktonic and biofilm bacteria to gentamicin [21]. Cinnamaldehyde was a biofilm inhibitor of *Pseudomonas aeruginosa*, and it appears to work synergistically with colistin. Resveratrol is a stilbene phytoantigen widely distributed in grapes, which has the characteristics of anti-QS and antibiofilm against *E. coli*. It can increase the effectiveness of different antibiotics against bacteria [22,23,24].

QS is a way for bacteria to coordinate population communication. It can regulate biofilm formation, virulence factor production, motility, and antibiotic resistance [25,26,27,28]. In Gram-negative and Gram-positive bacteria, AI-2 is considered to be a “universal” QS signaling molecule involved in inter- and intra-species communication. For most bacteria, the AI-2 QS system can regulate the virulence production, biofilm formation, and motility [29,30,31,32]. In *E. coli*, *pfs* and *luxS* are responsible for synthesizing AI-2 through the activated methyl cycle [33]. Together with other components of the *lsr* transport system, *lsrB* acts as a substrate-binding protein in the *ABC* transport system and plays a role in AI-2 internalization. *Lsrk* is a putative kinase that phosphorylates DPD to phospho-DPD. Moreover, phospho-AI-2 binds to transcriptional repressor *lsrR* to induce transcription of the *lsr* gene [34].

Biosynthesis of curli fimbriae and cellulose, two factors important for bacterial adhesion and biofilm formation, is controlled by *csgD*, a positive regulator of *csgAB* [35,36,37,38]. For instance, inhibition of curli-dependent biofilm formation by myricetin derivatives was shown by decreases in the transcription of *csgA* and *csgD* [39]. Bacterial flagellum functions as a rotary motor dependent on electrochemical differences between specific ions across the cytoplasmic membrane, which is closely related to motility [40,41]. *flhDC* is an activator of the flagella regulatory cascade, the second set of transcription factors for flagellar, and is related to the synthesis of early gene products of flagella [42,43]. *fliC* is the major subunit of flagella filaments responsible for *E. coli* motility [44]. *fliM* is the flagellar components of the C ring, components of flagellar switch, enabling rotation and determining its direction [45].

*E. coli*, a common opportunistic Gram-negative intestinal pathogenic bacterium for humans and animals, often causes diarrhea, acute enteritis, sepsis, urinary tract infection, and a series of diseases, and is still a significant cause of intestinal illness in humans and animals [46,47]. At the same time, *E. coli* is also a common pathogen of biofilm infection [6]. It can not only adhere to medical devices (e.g., indwelling catheter, endotracheal intubation, etc.) to form chronic infection but also can form biofilms of the bladder and intestine, leading to persistent and chronic infections, causing a significant threat to livestock and poultry breeding and human health [46,48,49].

In response to these challenges caused by *E. coli*, the current study aims to use antivirulence therapy that reduces pathogenicity alone without any harmful impact on bacteria [50,51,52,53]. In addition, compounds with non-bactericidal properties have effects on reducing the likelihood of developing drug resistance and mutants [54].

In this context, it is reported that plant-derived compounds with a huge structural variety offer a wide range of benefits in handling infections [55,56,57,58]. Gin, a flavonoid isolated from the leaves of Ginkgo biloba, had antiinflammatory, antioxidant, antitumor, and other biological activities [59]. Despite many pharmacologic investigations, there have been no reports on the antibiofilm activity of Gin. Herein, this study unraveled the antibiofilm and antivirulence properties of Gin against *E. coli*, investigated the possible mechanism of Gin, and evaluated the synergistic effects of combining Gin with antibiotics.

## 2. Results

### 2.1. Effects of Gin on Growth and Metabolic Activity of E. coli

The effect of Gin on growth and metabolic activity of *E. coli* was measured by microbroth dilution and Alamar Blue (AB) assay. Gin did not significantly adversely affect the growth and metabolic activity of cells in the range of 6.25~100 μM compared to the control group (Figure 1). These results revealed the non-bactericidal nature and unaltered metabolic activity of Gin on *E. coli*.

### 2.2. Inhibition of Biofilm Formation in E. coli by Gin

Different concentrations of Gin (6.25, 12.5, 25, 50, 100, 200, and 400 μM) were measured for antibiofilm activity by CV staining. This study showed that Gin treatment significantly inhibited the formation of *E. coli* ATCC 25922 in a dose-dependent manner (Figure 2A, *p* < 0.0001). As low as 6.25 μM of Gin was also found to be effective against the formation of biofilm (inhibition rate > 20%). To further observe the biofilm inhibitory effect of Gin, CLSM imaging analysis was conducted using fluorescent dye SYTO9 and propidium iodide (PI). The result showed that Gin significantly inhibited biofilm formation, as demonstrated by a negative gradient in green fluorescence in *E. coli* ATCC 25922 strains (Figure 2B). In short, the results showed that Gin inhibited biofilm formation.

### 2.3. Cytotoxicity of Gin on Caco-2 and IPEC-J2 Cells

Next, we examined whether Gin showed cytotoxicity toward Caco-2 and IPEC-J2 cells (Figure 3). Gin showed no cytotoxic effect at 50 µM or lower concentrations, and cytotoxicity appeared at 100 µM and 200 µM (*p* < 0.05 and *p* < 0.0001, respectively). These results indicate that Gin was not cytotoxic at concentrations effective in inhibiting biofilm formation.

### 2.4. Effects of Gin on EPS Production of E. coli

In biofilms, bacteria produce *EPS* that help entrap nutrients and serve as defense mechanisms [60]. *EPS* production in biofilms was studied using Ruthenium Red staining. Gin inhibited *EPS* production in a dose-dependent manner (Figure 4, *p* < 0.01, and *p* < 0.0001). As low as 6.25 μM of Gin was also found to be effective against the production of *EPS* (inhibition rate > 20%). This result was consistent with the inhibitory effect of Gin on *E. coli* biofilm. The results showed that Gin reduces the *EPS* production of *E. coli*.

### 2.5. Effects of Gin on the Motility of E. coli

Due to the closely related relationship between biofilm formation and bacterial motility, Gin was evaluated for its effect on motility and antibiofilm activity. As seen in Figure 5A, Gin at different concentrations significantly inhibited *E. coli* motility compared with a control group (*p* < 0.0001). Further, the size of the halo zone was quantitatively determined (Figure 5B). The halo zone of Gin with different concentrations was significantly smaller compared with the control group (*p* < 0.0001). This result proved that Gin could inhibit the production of biofilm by inhibiting the motility of *E. coli*.

### 2.6. Effect of Gin on AI-2 Production in E. coli

Biofilm formation by bacteria depends on QS [61]. Further, the bioluminescence assay of *V**. harveyi* BB170 was used to determine whether Gin could affect AI-2 production of QS induced by *E. coli*. As shown in Figure 6, the inhibitory effect of Gin on bioluminescence was dose-dependent, with an IC_50_ value of 22.33 μM. Our data indicated that Gin inhibited the formation of biofilm by inhibiting the production of AI-2.

### 2.7. Effect of Gin on the Transcription of Biofilm-Regulated Genes of E. coli

To elucidate how Gin might inhibit the formation of biofilms, we assessed the transcription of genes such as curli-regulated genes (*csgA*, *csgD*) and flagella-formation genes (*flhC*, *flhD*, *fliC*, *fliM*), and QS-related genes (*luxS*, *lsrB*, *lsrK*, *lsrR*) by qRT-PCR. As shown in Figure 7, Gin significantly inhibited curli-regulated genes *csgA*, and *csgD* (82%, 58%, respectively), flagella-formation genes *flhC*, *flhD*, *fliC*, and *fliM* (55%, 53%, 52%, and 79%, respectively) and QS-related genes *luxS*, *lsrB*, *lsrK*, and *lsrR* (44%, 60%, 56%, and 51%, respectively). These results demonstrated that Gin inhibited the formation of biofilm by inhibiting the transcription of the curli genes, flagella genes, and QS genes of *E. coli*.

### 2.8. Synergistic Effects of Gin in Combination with Antibiotics against Six E. coli Strains 

Biofilm bacteria are known to be able to increase their antibiotic resistance. Therefore, it is urgently important to find a way to reduce biofilm formation by overcoming these increasingly serious infections. Here, by combining marketed antibiotics, including gentamicin and colistin, with the antibiofilm compound Gin, we can combat *E. coli* biofilms. The synergistic effects of Gin (50 µM) combined with antibiotics on *E. coli* ATCC 25922 and five isolated clinical strains (*E. coli* XJ24, *E. coli* O101, *E. coli* O149, *E. coli* KD-13-1, and *E. coli* C83654) were investigated. The MIC of gentamicin, colistin B, and colistin E against *E. coli* ATCC 25922, XJ24, O101, O149, and KD-13-1 were tested (Appendix A). As depicted in Figure 8, Figure 9 and Figure 10, antibacterial activity revealed that combining gentamicin, colistin B, and colistin E (1/2 MIC, 1/4 MIC, 1/8 MIC) with antibiofilm inhibitor Gin on *E. coli* ATCC 25922 and five isolated clinical strains has a synergistic effect. Gin showed biofilm inhibition properties and significantly increased the antibacterial activities of gentamicin, colistin B, and colistin E. These results suggested that the antibiofilm compound Gin, in conjunction with conventional antibiotics, could be an efficient therapeutic strategy for tackling pathogens such as *E. coli*.

## 3. Discussion

Antivirulence therapy is an important defense mechanism for the host against pathogenic bacteria infection [62,63]. Here, the present study was the first to report the antibiofilm and antivirulence activity of Gin against *E. coli* without affecting its growth. Furthermore, Gin did not exhibit cytotoxicity within the effective concentration range in IPEC-J2 and Caco-2 cells, resulting in a non-toxic effect in humans. We also demonstrated that Gin exhibited synergistic effects when combined with existing antibiotics (e.g., gentamicin, colistin B, and colistin E). Thus, we concluded that Gin presents antibiofilm and synergistic effects against *E. coli*.

*EPS* forms pathways for transporting metabolites and nutrients, maintaining biofilm structure, and protecting biofilms from external pressure. In addition, the secretion of *EPS* promotes bacteria’s adhesion and colonization on nonbiological surfaces [64]. In our study, the effects of Gin on *EPS* production were dose-dependent, resulting in decreased bacterial colonization and biofilm formation. Additionally, it is reported that Gin effectively inhibited biomass of biofilm adhered to the cover slip surface by CLSM. These results showed that Gin could reduce biofilm formation by inhibiting the production of adhesive molecules [65,66].

The possible mechanism of Gin’s antibiofilm activity was identified using gene transcription analysis. The qRT-PCR showed that Gin significantly down-regulated the transcription of flagella-formation genes (*flhC*, *flhD*, *fliC*, *fliM*), curli-related genes (*csgA*, *csgD*), and QS-related genes (*luxS*, *lsrB*, *lsrK*, *lsrR*) (Figure 7). The curli, the main component of a complex extracellular matrix produced by *E. coli*, is regulated by *csgA*, and *csgD* [67]. Results showed that Gin significant down-regulated the transcription of *csgA*, and *csgD*, thus inhibiting the curli production, which corresponded with the results of *EPS* production (Figure 4). Additionally, results also showed a significant reduction in the transcription of *flhC*, *flhD*, *fliC*, and *fliM*, indicating that Gin significantly inhibited the transcription of flagella-regulated genes in *E. coli*, thus reducing the motility of *E. coli*, which corresponded with the results obtained in semi-solid agar medium for the motility assay (Figure 5). Furthermore, to assess whether Gin affects QS, we verified the effect of Gin on the transcription of *luxS*, *lsrB*, *lsrK*, and *lsrR*. AI-2 was a type II QS signal molecule in *E. coli*. It can affect phenotypes of *E. coli*, including biofilm formation, motility, virulence, bacterial adhesion, and biofilm matrix production [68,69,70,71,72]. Results showed that Gin reduces the transcription of *luxS*, *lsrB*, *lsrK*, and *lsrR*, which was consistent with the results obtained in the AI-2 bioluminescence assay (Figure 6). Combined with all these findings above, biofilm inhibition by Gin may be due to inhibiting QS and then regulating motility and the *EPS* production of *E. coli*. However, further research is needed to determine the exact mechanism of action.

Biofilm is a structured bacterial community, which poses a significant challenge to traditional antibiotics’ effectiveness and is considered a breeding ground for antibiotic resistance. Combined with marketed antibiotics, biofilm inhibitors can enhance bacterial sensitivities to antibiotics [73]. The present study also proved that antibiofilm inhibitors could enhance the sensitivity of traditional antibiotics to resistant strains (Figure 8, Figure 9 and Figure 10). The results suggested that combining antibiotics with antibiofilm compounds appears to be an effective therapeutic strategy for treating pathogen infections.

## 4. Materials and Methods

### 4.1. Materials, Bacterial Strains, and Cells

Gin used in this study was obtained from Shanghai Yuanye Bio-Technology Co., Ltd. (Shanghai, China). Based on the water solubility of Gin, it was dissolved in dimethyl sulfoxide (DMSO) at a final concentration was 40 mM, and the concentration of DMSO in LB medium was not more than 1%.

*E. coli* ATCC 25922 was obtained from American Type Culture Collection (ATCC). Clinical isolate strains *E. coli* O101, *E. coli* O149, *E. coli* C83654, *E. coli* XJ24, and *E. coli* KD-13-1 were isolated and maintained in our laboratory. Luria-Bertani (LB, HuanKai Microbial, Guangdong, China) and Luria-Bertani agar (LA, HuanKai Microbial, Guangdong, China) medium were used for the growth of all *E. coli* strains. *Vibrio harveyi* BB170 (*V. harveyi* BB170) and *V. harveyi* BB152 were gifts from Researcher Han Xiangan (Shanghai Veterinary Research Institute, Chinese Academy of Agricultural Sciences), and were cultured under conditions containing AB medium supplemented with 1 mM L-arginine, 1% phosphate buffer (pH = 7.2), and 1% glycerol.

Caco-2 cell lines were obtained from ATCC and cultured under standard conditions using MEM medium with 20% FBS, 1 mM sodium pyruvate, 1 mM L-glutamine, 10 mM HEPES, and 1% non-essential amino acids. IPEC-J2 cell lines were obtained from Beina Chuanglian Biology Research Institute and cultured under standard conditions using DMEM medium with 10% FBS, 1 mM sodium pyruvate, and 1 mM L-glutamine, 10 mM HEPES, and 1% non-essential amino acids.

### 4.2. Growth and Metabolic Activity

A microbroth dilution test was conducted to determine the growth effects of Gin by the Clinical & Laboratory Standards Institute (CLSI) [74]. In brief, cell suspensions of *E. coli* (10^6^ CFU/mL) and different concentrations of Gin were incubated for 24 h at 3 °C. A Multiskan Go Reader (Thermo Fisher Scientific, USA) was used to measure the absorbance at 600 nm following incubation.

In continuation, *E. coli* metabolic activity was analyzed by the Alamar Blue (AB) assay according to the previous method [75]. Briefly, cells were collected from each well using fresh 2 mL tubes followed by centrifugation for 10 min at 5000 rpm, washed twice with PBS (pH = 7.2), and then resuspended in 1 mL of PBS (pH = 7.2). Next, the tubes were incubated in dark conditions for 1 h at 37 °C, with 10 µL of AB dye (Invitrogen™, Thermo Fisher Scientific, USA) added. The blank control was PBS (pH = 7.2) containing only AB dye. The metabolic activity was calculated based on the absorbance at 570 nm and 600 nm using the following formula: $$Metabolic\;activity(\%)=\frac{\left(\left(E_{oxi\left(OD570\right)}\times T_{OD570}\right)-\left(E_{oxi\left(OD600\right)}\times T_{OD600}\right)\right)}{\left(\left(E_{red\left(OD570\right)}\times B_{OD570}\right)-\left(E_{red\left(OD600\right)}\times B_{OD600}\right)\right)}\times 100\%$$
*E_oxi (OD570)_*—extinction coefficient of AB in its oxidized form at 570 nm = 80,586;*E_red (OD570)_*—extinction coefficient of AB in its reduced form at 570 nm = 155,677;*E_oxi (OD600)_*—extinction coefficient of AB in its oxidized form at 600 nm = 117,216;*E_red (OD600)_*—extinction coefficient of AB in its reduced form at 570 nm = 14,652;*B*—blank; *T*—samples.

### 4.3. Biofilm Assay

#### 4.3.1. Crystal Violet (CV) Staining

The previous method was slightly modified to detect the formation of *E. coli* biofilm using CV staining [76]. Briefly, *E. coli* ATCC 25922 was statically grown for 24 h at 37 °C. Next, the cells were resuspended at OD_600_ of 1.0 and diluted about 100-fold. Subsequently, the diluted bacterial solution was mixed with Gin in a white 96-well plate (Corning Costar^®^ 3599, Corning, Kennebunk, ME, USA). Cells underwent static incubation for 24 h at 37 °C, were washed three times with PBS (pH = 7.2) to eliminate the nonadherent cells, and fixed for 1 h at 60 °C. After fixation with methanol and staining with 0.1% CV for 30 min, the stained biofilm was rinsed with tap water to remove the dye that was not bound. The CV contained in biofilm was dissolved in 150 µL of 95% ethanol, and its absorbance at 570 nm was measured.

#### 4.3.2. Confocal Laser Scanning Microscopy

According to a previous report, we analyzed the *E. coli* biofilms using confocal laser scanning microscopy (CLSM) [77]. In brief, *E. coli* (10^6^ CFU/mL) was inoculated in LB broth with or without Gin (0, 12.5, 25, and 50 µM) to form static biofilms on the cover slips for 24 h at 37℃ in the 12-well plates. The biofilms were washed twice with PBS (pH = 7.2) before removing loosely bound cells, and stained using a BacLight Live/Dead viability kit (L7012, Invitrogen™, Thermo Fisher Scientific, Eugene, OR, USA) according to the procedure. Afterward, excess staining was removed by washing twice with PBS (pH = 7.2), and then the biofilm was imaged by CLSM (LSM800, Zeiss, Jena, Germany).

### 4.4. Cytotoxicity

Gin cytotoxicity was assessed in Caco-2 and IPEC-J2 cells using the CCK-8 assay. Caco-2 and IPEC-J2 cells at a density of 1 × 10^5^ cells/mL were seeded into 96-well plates and cultured until the density of the cells reached approximately 80%. Then, the cells were treated for 24 h with 100 μL of Gin (6.25, 12.5, 25, 50, 100, 200 µM). Further, the cells with 10 μL of CCK-8 (MCE, Shanghai, China) were incubated for 1 h. After incubation, a Multiskan Go Reader (Thermo Fisher Scientific, Waltham, MA, USA) was used to measure absorbance at 450 nm.

### 4.5. EPS Production

The *EPS* production was quantitatively estimated by Ruthenium Red Staining [78]. Cell suspensions (100 µL) of *E. coli* (10^6^ CFU/mL) and different concentrations of Gin were cultured for 24 h at 37 °C. Afterward, the cells were washed with PBS (pH = 7.2). In each well, biofilm cells were stained with 0.01% ruthenium red (Yuanye, Shanghai, China) by incubation at 37 °C for 60 min. Ruthenium red was served as a blank. Following that, the solution containing the remaining stain was retransferred onto a new 96-well plate, and the absorbance at 450 nm was measured. *EPS* inhibition was calculated as:$$EPS\;\mathrm{inhibition}\left(\%\right)=\frac{\mathrm{AS}-\mathrm{AP}}{\mathrm{AB}-\mathrm{AP}}\times 100$$
where:AB = absorbance of blankAS = absorbance of sampleAP = absorbance of positive control

### 4.6. Motility Assay

Gin was evaluated for its effects on *E. coli* motility as described earlier [79]. In brief, *E. coli* cultures overnight were adjusted to an OD_600_ of 0.01. A semisolid agar media (0.3% LB agar) containing 12.5, 25, and 50 µM of Gin was used for the motility assay. A 1 µL measure of the diluted bacterial solution was inserted into the middle of the plate and then incubated for 24 h at 37 °C. The size of the halo zone compared to the control was used to evaluate the motility.

### 4.7. AI-2 Bioluminescence Assay

According to previous reports, AI-2 bioluminescence assays were performed [80,81,82]. In brief, incubation of *E. coli* (10^6^ CFU/mL) with 12.5, 25, or 50 µM of Gin for 16 h and centrifugation for 5 min at 12,000× *g* was performed. The supernatant was filtered with a 0.22 µm filter. The bioluminescence reporter *V. harveyi* BB170 grew in AB medium to OD_600_ of 1.0~1.1 and was then diluted with fresh medium at 1:2500. A 20 µL measure of cell-free supernatant was mixed with 180 µL of the *V. harveyi* BB170 dilution in black 96-well plates (Jingan, Shanghai, China) and incubated for 3.5 h at 37 °C. Measurement of bioluminescence was conducted using a multi-purpose microplate reader (Enspire, PerkinElmer, Waltham, MA, USA). Controls were cellular-free supernatants of overnight *V. harveyi* BB152 cultures.

### 4.8. qRT-PCR

qRT-PCR was used to study the effect of Gin on the transcription of biofilm-regulated genes of *E. coli*. *E. coli* was incubated with and without Gin for 24 h at 37 °C. Bacterial RNA Kit (Omega, Norcross, GA, USA) was used to extract total RNA. RNA quantity was measured by NanoDrop One^C^ spectrophotometer (Thermo Scientific, USA) and RNA quality was determined via agarose gel electrophoresis. Further, the RNA was reverse-transcribed into cDNA with a PrimeScript™ RT reagent Kit with gDNA Eraser (TAKARA Corporation, Kusatsu, Japan). The TB Green^®^ Premix Ex TaqTM II (Tli RNaseH Plus) (TAKARA Corporation, Kusatsu, Japan) was used for qRT-PCR. The 2−∆∆Ct method was used to assess relative changes in gene transcription levels. The *gapA* gene was used as an internal control [83]. In Appendix A, the primers used in the current study are listed.

### 4.9. Antibacterial Activity

Based on the previous method [75,84], we modified some procedures to evaluate the antibacterial effects of Gin. Briefly, after overnight culture, bacteria were diluted to an OD_600_ of 0.01. The diluted bacterial solution was mixed with antibiotics (1/2 MIC, 1/4 MIC, 1/8 MIC) with or without Gin (50 µM). The metabolic activity of mixed suspensions was analyzed with the Alamar Blue (AB) assay after incubation for 16–18 h at 37 °C. Tests were conducted in triplicate.

### 4.10. Statistical Analysis

Comparison between two groups was performed using multiple *t*-tests, while multiple comparisons were made using non-parametric one-way ANOVA in GraphPad Prism (GraphPad Prism 8.0.1, GraphPad, San Diego, CA, USA). Results from all experiments were presented as the mean ± SD of three replicates.

## 5. Conclusions

In summary, this study demonstrated that Gin has an antibiofilm and synergistic antibacterial effect against *E. coli*. These results showed that Gin could inhibit biofilm without affecting bacterial growth and metabolic activity. Besides, Gin inhibited the bacteria motility, *EPS* production, and QS of *E. coli*. More importantly, the antibacterial effects of combining gentamicin and colistin antibiotics with Gin were studied. The combination showed significantly synergistic effort against *E. coli* ATCC 25922 and five clinical isolates strains. In summary, these results indicate that Gin was an effective antibiofilm compound which can combine with marketed antibiotics to treat drug resistance caused by biofilms to treat *E. coli* infections in the future.

## Figures and Tables

**Figure 1 ijms-23-08809-f001:**
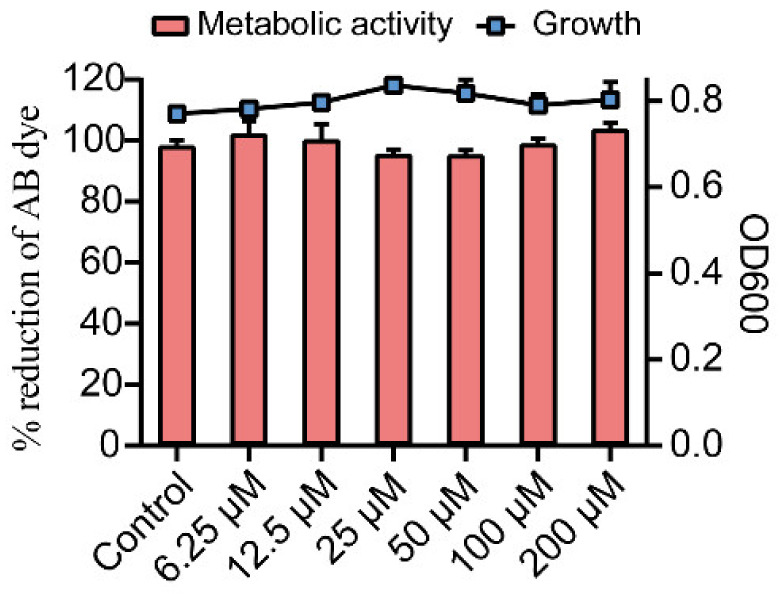
Growth and metabolic activity of *E. coli* in the presence of Gin. The line graph shows the growth of *E. coli* with Gin using a microbroth dilution assay. The bar graph shows the metabolic activity of *E. coli* based on the AB assay. *n* = 3. Results from all experiments are presented as the mean ± SD of three replicates.

**Figure 2 ijms-23-08809-f002:**
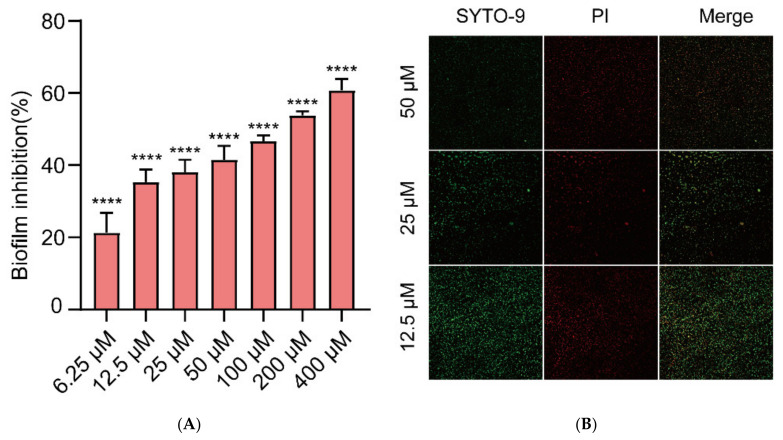
Gin’s effects on the growth of biofilms in *E. coli*. (**A**) Biofilm formation inhibition at various concentrations of Gin (6.25, 12.5, 25, 50, 100 200, and 400 μM) for 24 h. **** = *p* < 0.0001. (**B**) CLSM image of *E. coli* biofilm at different concentrations of Gin (12.5, 5, and 50 μM) for 24 h. *n* = 3. Results from all experiments are presented as the mean ± SD of three replicates.

**Figure 3 ijms-23-08809-f003:**
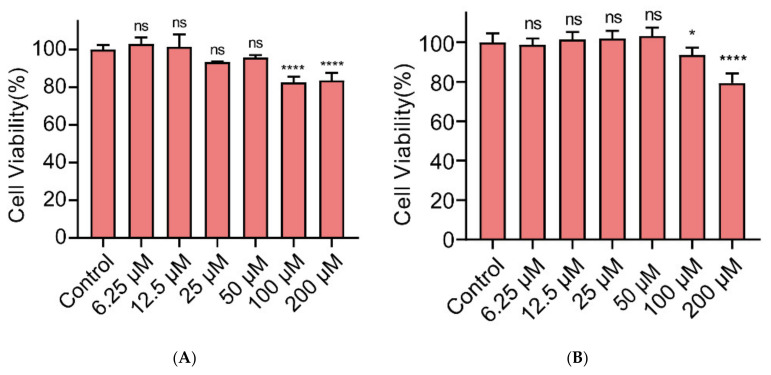
The cytotoxicity of Gin in (**A**) Caco-2 cell and (**B**) IPEC-J2 cell. Gin was applied to cells at various concentrations (0, 6.25, 12.5, 25, 50, 100, and 200 µM) for 24 h. *n* = 3. Results from all experiments are presented as the mean ± SD of three replicates. ns: no significant, * = *p* < 0.05, **** = *p* < 0.0001.

**Figure 4 ijms-23-08809-f004:**
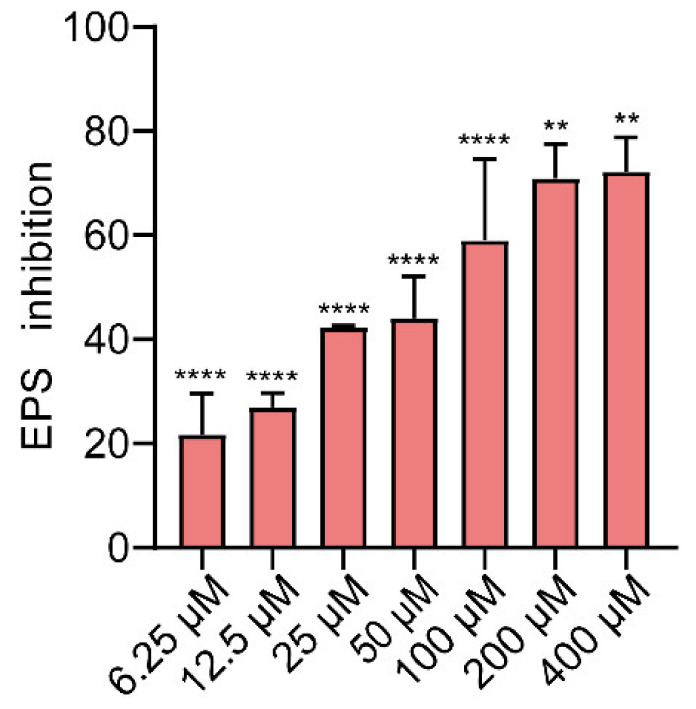
Results of *EPS* inhibition (%) at various concentrations of Gin (0, 6.25, 12.5, 25, 50, 100, 200, and 400 µM) for 24 h. *n* = 3. Results from all experiments are presented as the mean ± SD of three replicates. ** = *p* < 0.01, **** = *p* < 0.0001.

**Figure 5 ijms-23-08809-f005:**
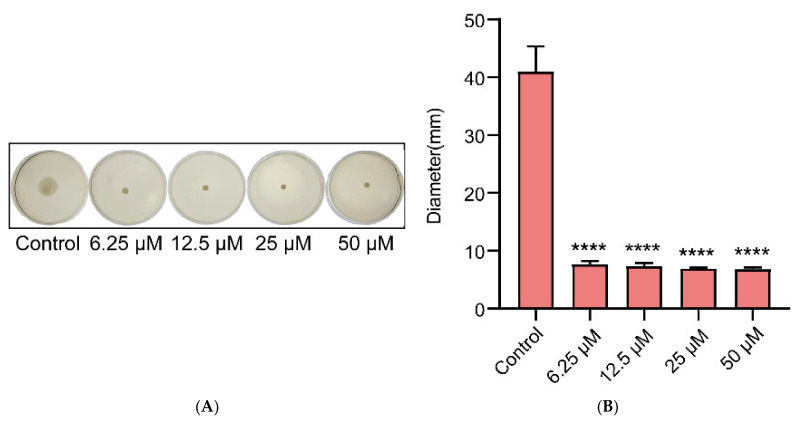
Motility inhibition of *E. coli* with Gin. (**A**) Images of motility following incubation with *E. coli* at various concentrations of Gin (0, 6.25, 12.5, 25, and 50 µM). (**B**) Quantitative estimation of motility based on the diameter of the halo zone. *n* = 3. Results from all experiments are presented as the mean ± SD of three replicates. **** = *p* < 0.0001.

**Figure 6 ijms-23-08809-f006:**
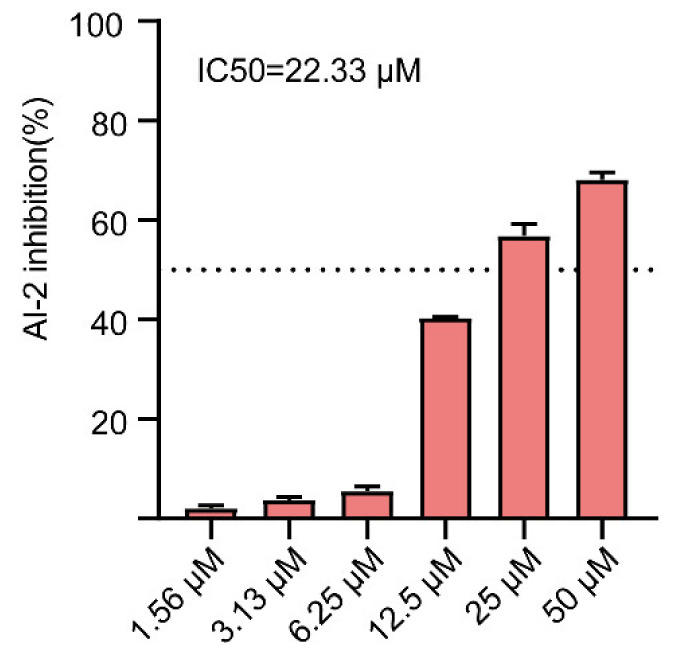
Inhibition of AI-2 activity treated with different concentrations of Gin (1.56, 3.13, 6.25, 12.5, 25, and 50 µM). *n* = 3. Results from all experiments are presented as the mean ± SD of three replicates.

**Figure 7 ijms-23-08809-f007:**
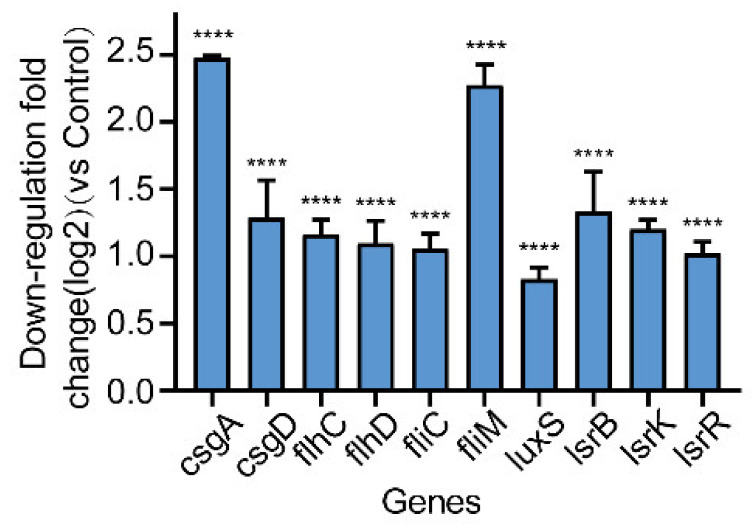
Effects of Gin on transcription of biofilm-regulated genes. The qRT-PCR results revealed significant difference in ten genes (*csgA*, *csgD*, *flhC*, *flhD*, *fliC*, *fliM*, *luxS*, *lsrB*, *lsrK*, and *lsrR*) as compared with the control. Results from all experiments are presented as the mean ± SD of three replicates. **** = *p* < 0.0001.

**Figure 8 ijms-23-08809-f008:**
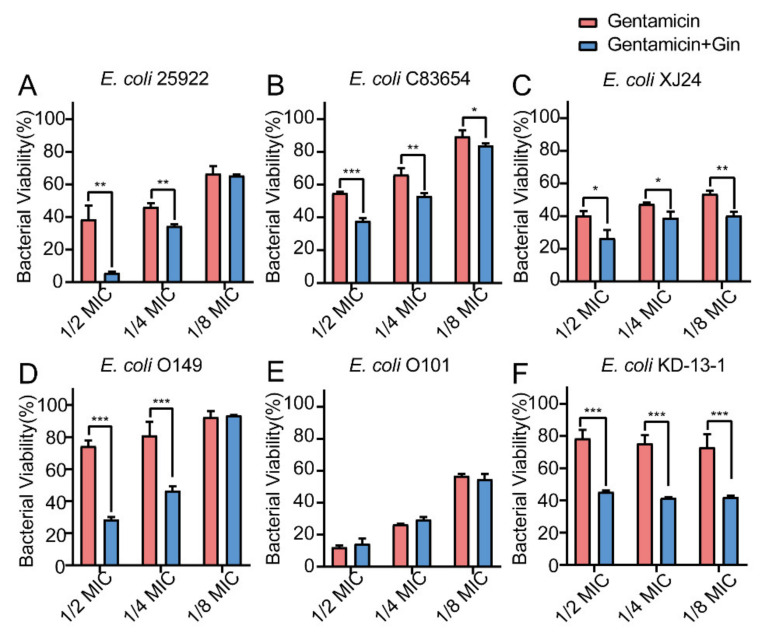
Effects of Gin (50 μM) and Gentamicin on bacterial viability in six different *E. coli* strains. (**A**) ATCC 25922, (**B**) C83654, (**C**) XJ24, (**D**) O149, (**E**) O101, (**F**) KD-13-1. *n* = 3. Results from all experiments are presented as the mean ± SD of three replicates. * = *p* < 0.05, ** = *p* < 0.01, *** = *p* < 0.001.

**Figure 9 ijms-23-08809-f009:**
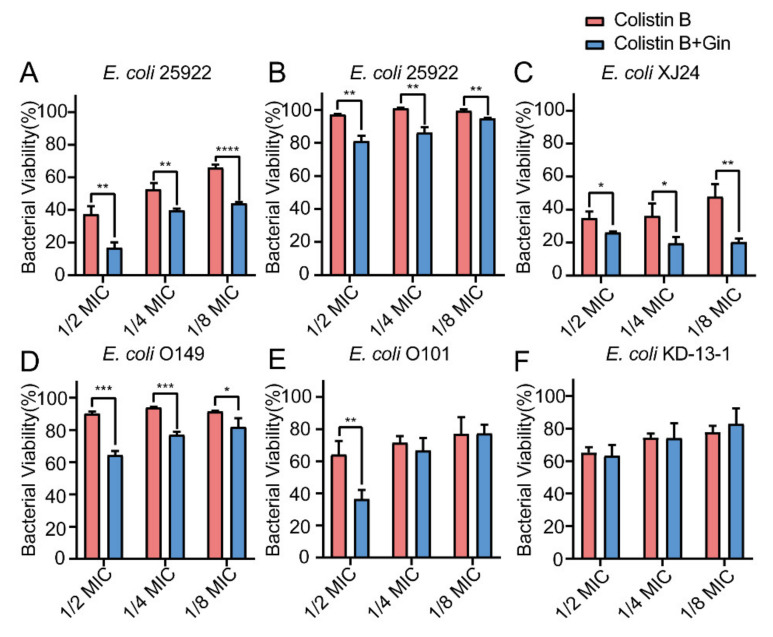
Effects of Gin (50 μM) and Colistin B on bacterial viability in six different *E. coli* strains. (**A**) ATCC 25922, (**B**) C83654, (**C**) XJ24, (**D**) O149, (**E**) O101, (**F**) KD-13-1. *n* = 3. Results from all experiments are presented as the mean ± SD of three replicates. * = *p* < 0.05, ** = *p* < 0.01, *** = *p* < 0.001, **** = *p* < 0.0001.

**Figure 10 ijms-23-08809-f010:**
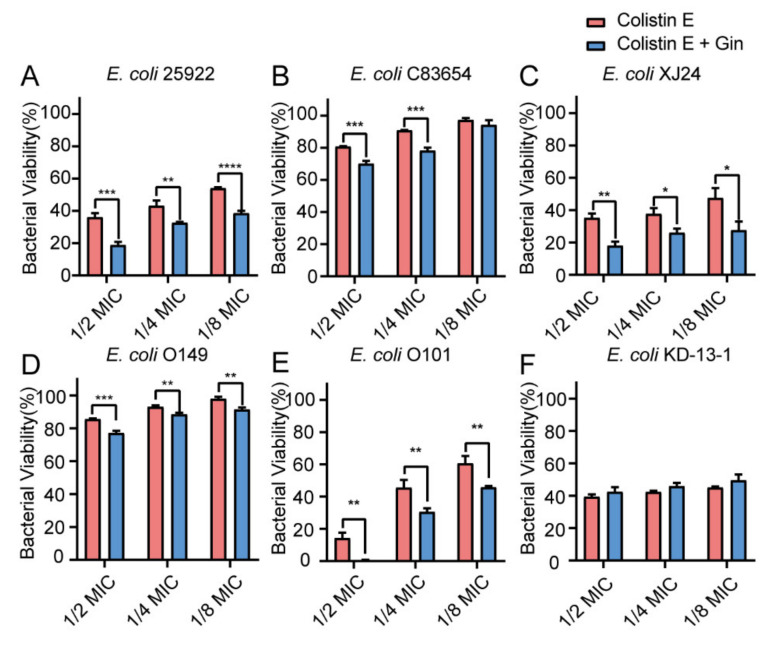
Effects of Gin (50 μM) and Colistin E on bacterial viability in six different *E. coli* strains. (**A**) ATCC 25922, (**B**) C83654, (**C**) XJ24, (**D**) O149, (**E**) O101, (**F**) KD-13-1. *n* = 3. Results from all experiments are presented as the mean ± SD of three replicates. * = *p* < 0.05, ** = *p* < 0.01, *** = *p* < 0.001, **** = *p* < 0.0001.

## Data Availability

Not applicable.

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
