# Peer review of "Novel Antibiofilm Inhibitor Ginkgetin as an Antibacterial Synergist against Escherichia coli"

_ijms, 2022, doi:10.3390/ijms23158809_

Round 1

Reviewer 1 Report

The article addresses the investigation of a molecule extracted from Ginkgo biloba that seems to affect biofilm production by Escherichia coli. The manuscript is very interesting and well structured. The discussion section should be implemented and the section on statistical analysis needs to be clarified.

Minor comment:

·         Page 1 line 37, bibliographic reference missing, please add it.

·         Page 2 line 55, please add the characteristics of the bacterium, e.g. that it is gram-negative, etc.

·         sub-chapter titles, as they are written in italics “E. coli” should not be written in italics.

·         Figures 2, 3, and 4, has the statistical analysis been carried out? please indicate the results. It is stated in the text that they are significantly different but the p-value is not indicated.

·         Page 4 lines 122-123, please indicate the p-value.

·         Page 4 line 132, what does QS mean? please clarify.

·         Page 5 line 145, I believe the authors were referring to figure 7, please check and correct if necessary.

·         Pages 7, and 8, lines 206-217, 225-229, 235-245, these parts seem to belong to the introduction section. Please move or edit them.

·         Page 8 line 252, by what criterion was the 40mM concentration chosen for Gin? Please clarify this in the text.

·         Page 9, the PBS had what concentration? Please specify in the text.

·         Page 11, Before applying the ANOVA test, was it verified that the data were normally distributed? If they were not, a non-parametric test should be applied instead of the ANOVA. Please specify and if necessary carry out the correct statistical analysis.

Reviewer 2 Report

A very interesting manuscript on the use of Ginkgetin as antibiofilm inhibitor and antibacterial synergist against E. coli. The manuscript is in general well written; however, attention is needed in some cases:

l. 76-77, 88-89 sentences seem truncated, please verify

l. 93, 95, 136, 148 please replace ‘inhibits’ with ‘inhibited’

l. 104 please consider replacing ‘cytotoxicity’ with ‘cytotoxic’

l. 142, 150 and elsewhere in text; since the mRNA levels were assessed, word ‘transcription’ seems more accurate than word ‘expression’.

Figure 7. please consider presenting the log2 values of the fold change. This will enable presentation of the reduction of mRNA levels in linear scale and may facilitate understanding.

l. 172, 177, 182 please replace word ‘species’ with ‘strains’

l. 213, 215, 216, 228 please pay attention to scientific nomenclature; genes are written in italics with first letter not capitalized while proteins are not written in italics with the first letter capitalized. In these lines, this is not clear

paragraph 4.8. how was RNA stabilized?

l. 355 ‘gapA’ should be written in italics

l. 356 table S2

l. 464 ‘Elletaria cardamomum’ should be written in italics
